# How Rodent Species Adapt to the Food Resources of Their Habitat

**DOI:** 10.3390/ani15131874

**Published:** 2025-06-25

**Authors:** Sergio Del Arco, Jose María Del Arco

**Affiliations:** 1Faculty of Biology, University of Salamanca, Campus Miguel de Unamuno, 37007 Salamanca, Spain; sergiodelarco@usal.es; 2Department of Agroforestry Sciences, Higher Technical School of Agrarian Engineering of Palencia, 34004 Palencia, Spain; 3iuFOR Sustainable Forest Management Research Institute, University of Valladolid, 47002 Valladolid, Spain

**Keywords:** acorns, dispersal, rodents, scatter-hoarding, partial consumption, habitats

## Abstract

Relationships between rodents and oak species have been interpreted in terms of the role of the rodents. These relationships were initially characterised by predation by the rodents. However, over time, the relationships have begun to move towards collaboration. This work examines the behaviour of three rodent species when handling the seeds of two oak species. Depending on how long the relationship has been maintained, the rodent species acquire certain adaptations that approximate the relationship to one of collaboration, favouring the survival of the oak species. We have investigated which rodent species participate in the acorn dissemination process, encouraging the establishment of oak species. We have also examined how three rodent species may coexist in the same habitat without excluding one another, despite the strong competition for resources. It is found that they separate the habitats occupied by each specific species.

## 1. Introduction

Currently, three rodent species with very similar characteristics (Wood mouse, Algerian mouse and Common vole) coexist in the central plateau of the Iberian Peninsula. Two of these have been using the resources and adapting to the area’s conditions for a long time (Wood mouse, Algerian mouse), but the other species has only recently arrived (Common vole) [1,2,3]. By comparing their behaviour regarding resource use and habitat occupation, we attempt to learn how they have adapted to the conditions of the occupied environment. To coexist, we believe that these three rodent species with similar characteristics should segregate their niches and habitats to prevent the intense competition for resources from eliminating any of the competing species, as established by the principle of competitive exclusion [4]. The adaptations acquired over time by each species in this respect should appear in the experiment that has been carried out. In this work, we compared the behaviour of the three rodent species occupying the three habitat types and their behaviour with respect to the processing of one of the most abundant resources of the area: acorns, the seeds of the oak species. Over recent years, the relationship between rodent species and the oak has attracted the attention of several research groups [5,6,7,8,9,10,11]. However, in many of their works, the interpretation of the outcome of these relationships has changed depending on the pairs of species studied. In some cases, the rodents participate and collaborate in seed dispersal, but in others, they only act as predators [12].

From an ecological point of view, relationships between species are studied in terms of the benefits or detriments obtained by each species during the relationship [13]. The outcome of the relationship for each species may be beneficial if the target species gains individuals or resources. It is detrimental if the species loses individuals or resources. If there is no change in the number of individuals or resources available to the species, it is considered neutral. Based on these premises, several types of relationships have been defined: Competition, when both species involved suffer detriment; amensalism, when one suffers detriment and the other is undisturbed; commensalism, when one species benefits and the other is unaltered; predation, when the predator species benefits and the prey species is harmed; neutralism, when neither species experiences variation; and mutualism, when both species benefit. By observing these relationships in various types of species, however, it has been found that the relationship can change over time. For example, an asymmetrical competitive relationship in which one species suffers greater harm than the other may end up as amensalism. If the prey species does not suffer a significant depletion of its resources, predation may turn into commensalism, as is the case with the galls of hymenopterans and the oak species.

This research follows in this direction. Our aim is to test whether acorn predation by rodents may be transformed into a relationship more akin to mutualism. Some authors have found that the relationship between scatter-hoarding rodents and oak species varies in their outcomes [6,7,8,9,10,11]. At one end of the range is antagonism, a predation relationship in which rodent species take resources from oaks, which may suffer costs [12]. At the other end is mutualism, in which both species derive benefits. In mutualism, the benefits must be symmetrical for both species. Therefore, relationships between rodent and oak species will never be considered mutualism since the rodent always obtains more benefits than the plant. However, the contribution of some rodent species with the transport and storage of acorns to the oak dispersal process brings the relationship closer to mutualism and separates it from predation [14,15,16].

A mathematical model has been proposed [10,17,18] to assess whether the outcome of the scatter-hoarding relationship is antagonism or mutualism [19,20]. When the benefits of transportation and burial are greater than the costs incurred by predation during this scatter-hoarding process, the relationship may be considered mutualistic.

This range of results in the rodent-oak relationships leads us to believe that the observed variations may be a result of the duration of the relationships. Of the three species investigated in this study, the two that have been consuming acorns for the longest time may have acquired adaptations that the third, more recently arrived species, has been unable to acquire due to a lack of time.

In previous studies we have found that the Common vole preys on acorns when it uses them as food, as it opens them at the apical end, eating the embryo first. We have also found that it does not bury acorns for storage. Recently personal communications have prompted us to think that possibly this species is starting to use acorns as a source of nutrients. The questions we pose are: Is this species changing its acorn utilisation behaviour, is it occupying the acorn-producing areas, the oak woodlands, has it changed its acorn hoarding behaviour?

Our aims were to (1a) Test whether the duration of the rodent-oak relationship modifies the relationship through the adaptations that are acquired by each species. We question whether the amount of time elapsed since the start of acorn consumption makes a difference in the acorn treatment during consumption. To test this, we compared the behaviour of three rodent species that spent distinct amounts of time handling acorns.

(1b) Within this objective, we would like to know if the Common vole has modified its way of dealing with acorns during the time it is occupying the northern plateau, as it seems that it is starting to use acorns as food.

(2a) Test whether habitat segregation exists between rodent species. This would permit coexistence without unduly increasing competition. Are there habitat preferences between rodent species? To verify this, we studied the occupation of different habitats by the three rodent species.

(2b) Especially within this objective, we will be able to estimate whether the Common vole is starting to occupy the oak woodlands if it is active in the scrub habitat.

(3a) Know the degree of involvement of each rodent species in the process of dissemination of oak acorns through the transport and storage of acorns. Does each rodent species transport and store acorns? Where do they store them? To determine this, we tracked the fate of the acorns after their processing by the rodents.

(3b) As a subsection within this objective, we can estimate whether the Common vole has changed its acorn-hoard behaviour: Does it perform caching?

## 2. Materials and Methods

### 2.1. Experimental Procedures and Design

For this study, we captured, by means of Sherman traps, three males and six females specimens of the three mentioned rodent species from the study area: Wood mouse, Algerian mouse and Common vole. We placed one male and two females in three farrowing crates in the laboratory per species [21]. From their offspring, we selected seven specimens per species, having a similar age (four weeks). The purpose of this was to obtain similar specimens, having homogeneous characteristics, to ensure that no biases existed that may influence the result. None of the specimens had prior contact with acorns. This allows us to determine whether, over time, variations occur in the acorn-handling behaviour. This would suggest that a learning period exists regarding acorn handling. If no differences were found, this would imply that the behaviour is linked to inheritance and is typical of the species.

Another objective of this procedure was to avoid aggressive confrontations between specimens of the same species in the same plot, which may lead to the death and disappearance of a specimen [2,21,22].

We selected three 100 m^2^ plots from an oak forest dominated by *Quercus ilex* and *Quercus faginea* species. We isolated these plots to prevent the entry of other animals. To prevent the rodents from exiting the plot, the perimeter was covered with 2 m high metal sheets, buried 50 cm into the ground and extending 1.5 m above the surface. This kept the Wood mouse species from jumping out. In each of these plots, one family of each rodent species was placed, according to the described composition of members.

Within each of the three isolated plots, three longitudinal strips measuring 3 m in width were constructed, positioned 0.5 m apart from one another. They were manipulated to create three separate environments (habitats). In the first, all vegetation was removed, and medium-sized stones (30 × 30 × 30 cm) were introduced. We call this habitat STONES. The intermediate strip was not manipulated. It consisted naturally of the existing vegetation, 30 cm to 2 m thickets and 2 to 4 m trees. This habitat we call SCRUB. The third strip only contained grass. We refer to this habitat as GRASS.

At one end of each strip, we placed a cylindrical cage (20 × 30 cm) with 4 openings opposite each other on the sides but close to the base, allowing the mice to enter and exit. The 4 opposing openings are for mice to access from any direction. In each cage, we placed a mixture of 150 *Q. ilex* acorns and 150 *Q. pyrenaica* acorns. We chose these two oak species to feed the rodents because they are the most abundant in the study area, and, therefore, rodent species that eat acorns are more likely to use these oak species as food. The forest in which we carried out the experiment is dominated by *Q. ilex,* accompanied by less abundant specimens of *Q. faginea*. However, as this species is less abundant in the region than *Q. pyrenaica*, we chose to include the latter in the experiment.

In each strip, 5 cylindrical plastic containers (15 cm diameter × 10 cm high) were introduced, buried at a depth of 15 cm. They were connected to the outside by a rubber tube measuring 3 cm in diameter and 25 cm in length to imitate the larder that is created by some of these rodent species in nature [23].

All acorns used in the experiment were labelled with a plastic tag attached to the acorn by a thin wire inserted into the cotyledons in the middle of the acorn. Each tag was numbered to identify each acorn, and we used different colours to identify the rodent species, oak species and habitat of origin. This made it possible to know the provenance of each acorn when they were recovered after being mobilised. Each day from the start of the experiment, sampling was carried out to recover all the acorns used each day.

Each tagged acorn was weighed before being fed to the mice. Once retrieved each day, after being mobilised, each acorn was reweighed with its tag and wire. This allowed the mass of each acorn consumed per day to be estimated.

Every day, the transported acorns were collected and classified according to their destination, depending on where they were found. These destinations were as follows: eaten in situ within the cage, abandoned on the ground surface, lightly buried in small caches of 4 or 5 acorns, or placed in the previously buried cylindrical containers (larders). The remains of the located acorns were weighed to calculate the mass consumed per acorn. As quickly as possible, the labelled acorn remains were returned to the site where they had been found, since these remains may be once again used as food by mice, as determined in [24].

The remains of the consumed acorns were classified into several categories according to how they were opened: opening at the basal end (B), opening at the apical end (A), fully eaten acorns (T) and intact, unconsumed acorns (I). In the acorns with an apical opening (A), we also noted whether or not the embryo was preserved.

By using the number of acorns processed in each habitat and, in parallel, the mass of acorns consumed in each habitat, we can check whether there are habitat preferences and answer the question of whether there is habitat segregation between species. By studying the number of acorns that each rodent species opens at one end or the other, consumes completely, or leaves intact, we will be able to answer the question of whether there are differences in the handling of acorns by rodent species. Do they have different adaptations in the handling of acorns? Finally, using the number of acorns that we find after they have been mobilised, we can check whether each rodent species participates in the dispersal of acorns. These are the three planted objectives.

The experiment was stopped in each species when any of the three acorn cages in each habitat were emptied. In the case of the Wood mouse, this occurred 16 days from the start, 21 days in the Algerian mouse and 32 days in the Common vole. The reason was that emptying acorns from a habitat could force rodents to shift their preferences to habitats with higher risk and distort the results.

### 2.2. Species Characteristics

The first of the three rodent species to reach the Iberian Peninsula was the Wood mouse. It has inhabited the peninsula since ancient times (Upper Pliocene) [25,26] and it currently occupies the entire peninsula.

The Algerian mouse has its origins in northern Africa, but it has been present in parts of the Iberian Peninsula since Neolithic times [2,26,27,28,29]. It currently occupies almost the entirety of the peninsula. It is excluded from the northern mountains that occupy the peninsula from west to east. This species prefers open habitats but is also found in agroecosystems such as crops, orchards, grasslands and scrubland, but not in forests [30]. In open habitats such as grasslands, it prefers tall grass that may be used to forge tunnels in which it shelters [21,31]. No mice have been found in mature forests, dense thickets or tree plantations [1]. Their low water requirements allow them to survive where other rodents may be eliminated. This species is part of the diet of over a dozen predators, including carnivores, owls and snakes [2].

The last of the three species to arrive on the central plateau of the Iberian Peninsula is the Common vole [3,32]. It arrived in the 1980s, coming from the mountains in the north of the peninsula. It had been confined to this area, which is the home of its regular food, fresh grass, which grows there thanks to the year-round rainy climate and mild temperatures. Because of the southward expansion of irrigated crops due to the construction of reservoirs, this species began to move, following the crops from the northern mountains to the southernmost steppe plateau.

Currently in the study area it (*Microtus arvalis*) does not consume them (acorns) because it has not come to occupy the oak forests that produce this seed. However, due to recurrent periods in which this species becomes a pest, devastating crops and pastures, its food disappears. In periods of prolonged drought, their usual food dries up and disappears as well. During these two periods of plague and drought, some specimens may seek new places and resources to feed. Thus, they may have reached the oak woodlands. This is the risk of the presence of this species moving from crops and meadows to these oak Woodland, as it feeds on the acorns, destroying the embryo, it only behaves as a predator and does not participate in the dissemination of acorns, it could drastically slow down the dispersal process. Consequently, this rodent is beginning to utilise a resource that was previously unavailable to it. According to some authors who work in the area with this species, they have begun to locate acorns in the tunnels and burrows of this species. However, these are personal reports that have not been published, so we cannot provide any documentation to support this. According to this, this species is beginning to occupy oak groves in this area, putting the dissemination of acorns at risk. For this reason, we have included this species in this experiment to check the use of scrub and oak habitat by this species (objective 2b), the management of acorns and its predatory behaviour (objective 1b) and to estimate if the risk for the dissemination of oak trees has already started (objective 3b). We cannot provide documents to prove these modifications, but depending on the results of this work, we will be able to check if the Common vole is starting to occupy the oak forests (scrub habitat) and if it is starting to consume acorns, as it has its usual food and acorns to choose from.

The three rodent species have different diets. The Common vole feeds on fresh green vegetables and is therefore confined to farmland or pastures created for domestic livestock. The Algerian mouse has a highly varied diet that includes all types of available food, ranging from a vegetable portion, including fruits and seeds such as acorns of various oak species, to insects and small invertebrates that complement its diet [28]. The Wood mouse is an omnivorous species, highly adaptable to the resources available at any given time. It mainly consumes berries, fruits and seeds, including acorns from oak species. At times, it also consumes live prey such as insects and their larvae.

The two oak species used in this work were selected because they are the most widespread in the study region [33]. The most abundant species in the area is the *Quercus ilex* [33]. It is the species that occupies the most territory and possibly has the greatest density, also being the most abundant. To reproduce, it produces acorns, a very nutritious food, given that 95% of its structure consists of cotyledons, which contain proteins, sugars and fats, substances that are greatly appreciated by all herbivores and omnivores. Another oak species that is also well represented in the study area is *Q. pyrenaica* [33]. Its extension is not as great as the previous oak, but it has similar densities. It also produces acorns for reproduction.

### 2.3. Data Analyses

The possible effects of rodent species, Quercus species, habitat and their interactions on the number of acorns and mass of acorns eaten per rodents were analysed using linear mixed models (LMM) with the Restricted Maximum Likelihood method (REML). The number and mass of the acorns were treated as random factors, and the time was considered a repeated factor. Finally, working on the model matrix, contrasts were performed to test differences between fixed factor levels [34]. Consequently, the Bonferroni correction was used to adjust for the significance level for each *t*-test [35]. Statistical calculations were implemented in the R software environment (version 2.15.3; Core Team R, Vienna, Austria, 2013), using the nlme package for LMM [36].

To examine the influence of rodent species, oak species and habitats on the mass of the consumed acorns, one- and two-way factorial ANOVAs were performed. In one of these, the influence of oak species and rodent species on the mass consumed was compared. In the other, the influence of rodent species and habitats on the mass of acorns consumed per day was compared. We tested the assumptions of normality by Shapiro-Wilk and homoscedasticity by Levene’s test.

## 3. Results

The results of the experiment were estimated as the number of acorns consumed daily by each rodent species in each habitat of both oak species (Table 1).

The distribution of the residuals revealed the good fit of the chosen model. We compared the fit with other models, but the AIC values indicated that the chosen model had the best fit.

Significant differences are found between rodent species in terms of the number of acorns consumed. The same differences are observed between the two oak species. More acorns are consumed from one species than from the other. Differences are also found in the number of acorns consumed between habitats.

The number of *Q. ilex* and *Q. pyrenaica* acorns consumed by each rodent species in the three habitats (Figure 1 and Figure 2) suggests that, in all three rodent species, there is a habitat that is not as appreciated. The number of acorns consumed in these habitats is very low, indicating that the rodent is not very active in these habitats.

Similarly, we have studied the interactions between habitats and species using the **mass** of acorns consumed per acorn per day by rodent species (Table 2).

The Algerian mouse, which is the smallest of the three species [37], ingests the lowest daily mass of each acorn. On the contrary, the largest species, the Common vole, consumes the highest mass per acorn per day (Table 3). Rodents generally consume more *Q. ilex* than *Q pyrenaica*. *Q. ilex* contains higher concentrations of sugars, proteins and fats [37], making it the more nutritious species. This may be why these acorns are preferred over those of the other species. Of the three rodent species, the Algerian mouse and the Wood mouse consume more of the *Q. ilex* acorns, but the Common vole does not prefer any of the two oak species.

To verify the acorn consumption rate of the three rodent species in the three habitats, we subtracted the number of acorns consumed or transported per day over the seven days of the experiment from the initial number of acorns of each species. We fitted a linear model to the values obtained and calculated the slope of the line to compare the speed of acorn transport by habitat and rodent species (Table 4).

Depending on the slope, the Wood mouse is the species that transports the acorns the fastest in the scrub habitat for the two oak species. The acorns of the two oak species are transported less quickly by the Algerian mouse in the rock and grass habitats. The Common vole also quickly transports acorns of both oak species in the grassy and rocky habitats. The Wood mouse displayed little activity in the grass habitat, and the Algerian mouse and Common vole do not spend much time in the scrub habitat.

To check the activity of the three rodent species in the three habitats, we estimated the mean mass of acorns consumed per acorn and day in the three species and the three habitats (Table 5). The mean value of the mass of acorns consumed in each habitat does not display significant differences. However, significant differences are seen when observing the mean consumption per acorn by each rodent species in each habitat. With this variable, the same trends are observed when using the slope of the straight line adjusted for the number of acorns disappeared per habitat. The same trends are also observed in the mass of acorns consumed by each rodent species. The Algerian mouse ingests the lowest mass, but with no difference between the three habitats. The mass consumed is slightly lower in the least visited habitat, shrubland. The Wood mouse and the Common vole do display significant differences in the habitat that is the least frequented by each.

To examine the treatment of acorns by each rodent species, we classified the remains of the acorns found after being transported into four different classes: those that were first eaten at the basal end (B); those that were first opened at the apical end (A), where the embryo is located; those that were eaten totally (T); and those that were not opened, which remained intact (I). Those that were first opened at the apical part may or may not retain the embryo (Table 6).

In the three columns on the left, we show the number of acorns of each class and the percentage that this number represents with respect to the total number of acorns provided (450 in the three habitats). The Apical class is divided into two parts since some of the acorns that were first opened at the apical end may retain the embryo after being opened. In the right-hand column, the number of basal plus intact and apical acorns that retain the embryo is added up to compile all acorns of these three classes that retain the embryo after consumption. We also added the percentage that this number represents with respect to the total number of acorns provided (450 in the three habitats). The two rodent species that process the most *Q. ilex* acorns are the Algerian mouse and the Wood mouse. A high percentage of the acorns that they process begin to be eaten at the basal end (33%). Together with the intact acorns that are not used, these acorns reach a percentage that is approximately 89% of the acorns retaining the embryo after being handled by these two rodent species. The behaviour of the Common vole is different. This species leaves many acorns intact, but it begins to open the acorns at the apical end, destroying the embryo.

The treatment of *Q. pyrenaica* acorns displays similar trends to those used with *Q. ilex* (Table 7), although in this case, the number of acorns attacked is lower since a higher number remain intact. The proportion of acorns that the Algerian mouse and Wood mouse open at the basal end remains around 30%. However, the Common vole opens the acorns at the apical end, destroying the embryo of almost all the acorns that it consumes.

In the two habitats where it is active, the Algerian mouse displays a preference for depositing the acorns of this oak species in the larder [23] (Table 8). It consumes a few acorns in situ, leaves a small number of acorns on the surface and stores many acorns in scattered caches. The amount stored in these caches is half the number that is deposited in the larder.

The contrary occurs with the Wood mouse (Table 8). This rodent species prefers to deposit *Q. ilex* acorns in scattered caches in the habitat that it occupies. It deposits even more in the stone habitat that it also visits, transporting the acorns that it deposits in caches made in the scrub habitat (its preferred habitat). However, it also transports acorns from the scrub habitat to the stone habitat, where it buries them in caches.

The Common vole consumes most of the acorns that it processes in situ. The number of acorns abandoned on the surface or introduced into larders is negligible [23]. It does not bury acorns in caches, refraining from engaging in this behaviour.

The same result was obtained for *Q. pyrenaica* acorns (Table 9). The Algerian mouse deposits more acorns of this oak species in the larder than in the other destinations. It buries acorns in caches, but only half as many are deposited there as compared to the larder. The Common vole also consumes acorns of this oak species in situ. The Wood mouse prefers burying acorns of this species in scattered caches, moving them between their preferred habitats.

## 4. Discussion

The three rodent species that have been chosen for this study share space in the central plateau of the Iberian Peninsula [3,28]. However, as we have verified, they segregate their habitats, perhaps to avoid engaging in intense competition for the same resources, which would be detrimental to all the competing species [4]. The Wood mouse was the first species to arrive in this region. It prefers to occupy the habitat consisting of thickets and small trees. This structure may be preferred since these thickets protect them from their predators [38]. Having occupied this region for a long time, it is likely to have had negative experiences with predators such as birds of prey [2]. These birds are not active in the thickets, which hinder their movements when searching for rodents. Therefore, it is possible that this rodent species seeks the protection of thickets for its feeding activities. It has been found [39] that this species is not present in the interior of large forests or monospecific formations. In these situations, the dominant woody species does not allow the proliferation of other species, even scrub. These tend to have homogeneous and unprotected surfaces where rodents would be stalked from tree branches without any protection. Rocks are another habitat that is often frequented by this rodent. Occupying the galleries between rocks or even moving around the base of rocks provides them with protection from predators. They are virtually inactive in grassy habitats. In these open areas, the rodent would be visible to birds of prey flying overhead. In this way, the Wood mouse separates its breeding habits from those of the Algerian mouse. This second species does not enter areas of dense scrub. It has occupied the Iberian Peninsula for less time than the Wood mouse, but it is attacked by the same birds of prey. It escapes from them by taking advantage of the galleries in the rocky habitat, which it occupies most intensively. It also moves through open crops and grassy habitats, where it could be attacked by predatory birds. However, it escapes from them by occupying tall grassy pastures. It uses this grass as a protective element, making galleries between it without digging into the ground. The tall grass covers the rodents from above and hides them from the view of predators [38]. As we can see, each species uses different strategies to deal with the same problem. The Common vole also inhabits crop and pasture areas where it is currently confined in the study region. Here it visits grassy habitats more intensively. It also suffers predation by birds of prey, but unlike the previous species, its escape strategy consists of building galleries, tunnels dug into the ground. This species also uses the stones habitat, where it finds galleries that have already been built in the proximity of the stones, thus escaping from predators. As can be seen, the Common vole is not active in scrub habitat. Therefore, we can now answer the important question: Is the Common vole beginning to occupy oak woodlands in response to drought and high populations? We can answer no. According to the results of this experiment, it has not yet entered this type of ecosystem.

This segregation of habitats inhabited by each of the rodent species allows them to coexist without competition for any specific resource, displacing or excluding any of the species [4].

The average mass of acorns ingested by each rodent species depends on its size. The smaller species consumes fewer acorns, while the larger species consumes a larger average mass of acorns, which is logical since it needs to maintain a larger body mass [40]. *Q. ilex* is the oak species with the highest mass consumed by the three rodent species, although in the Common vole, the difference does not reach significant levels. This difference in consumption may be because the *Q. ilex* species has higher concentrations of fats, sugars and proteins [37], giving it more nutritional properties. To cope with winter hardships, the *Q. ilex* acorns could provide greater reserves [41]. No significant differences exist in the amount of acorn mass consumed between habitats. These differences only appear in the habitats that were used the least by each rodent species.

The two oak species used here have been selected because they are the most widespread in the study region [33]. They are also the most abundant, perhaps because of their density. The Wood mouse is the first rodent species that arrived on the Iberian Peninsula during the Pleistocene epoch. During the Neolithic period, the Algerian mouse arrived from Africa, and the Common vole has only recently arrived in the northern plateau of the Iberian Peninsula from the northern mountains (1980s). The two rodent species that have been occupying the region the longest are likely to have encountered the acorns of the two oak species more frequently during their breeding season, given their abundance [42]. This is likely to be the reason why they have incorporated them into their diet, aside from the nutritional characteristics that their cotyledons possess [43]. The Common vole has only recently arrived and has not previously encountered this food since it does not occupy woodland or wooded areas. Its food consists of fresh green herbaceous plant material, which are the plants that it encounters in the agricultural areas and pastures where it lives [3]. In this experiment, its usual food was available in all three habitats, but more densely in the grassy habitat. This is probably the first time it has had acorns of these species to feed on. However, to the question of whether the Common vole is starting to consume acorns, we can answer that it probably is, as in this experiment they have their usual food and acorns. Observing the number of acorns and the mass consumed per acorn, we can see that in the habitats where it is active, it consumes a similar number and mass to the other two species. Therefore, it is beginning to appreciate acorns for their nutritional value and even prefers them to its usual food in the most used habitats. This would explain why it is possible that acorns are appearing in its burrows, as we have been told by colleagues. In the time that it has been occupying the northern plateau, it may be adapting to the consumption of a type of food that it did not previously incorporate into its diet. This behaviour poses a danger to the process of dissemination of acorns in oak species.

All these differences are reflected in the different ways of handling the acorns by each rodent species. The behaviour exhibited during the acorn processing differs among the three rodent species. The Wood mouse and the Algerian mouse have long consumed acorns. Both rodent species open most of the acorns that they process at the basal end, away from the position of the embryo at the apical end (between 24 and 33% of the available acorns and between 71 and 78% of the acorns attacked). This means that in a first attack, 71% of the acorns processed by these two rodent species retain the embryo and therefore, the ability to germinate. Why does this behaviour occur? This appears [44,45] to suggest a certain awareness of the rodent when engaging in this basal opening. However, this is not possible, since they spend more effort on this type of opening than on the apical opening. The apical end is narrower, and a smaller mouth opening is required. Rather, we believe that the rodents are forced to engage in this basal opening due to the characteristics of the acorns. According to [46,47,48,49], the oak species accumulate high concentrations of tannins in the embryo environment to prevent predators from eating the embryo. Based on the results of this experiment, we believe that the highest amount of tannins is concentrated in the apical shell surrounding the embryo. All acorns consumed in their entirety by the Wood mouse and the Algerian mouse retained one-third of the shell of the apical dome surrounding the intact embryo (Figure 3). This structure is probably not attacked by the rodents due to the bad taste produced by the tannins accumulated there. Thus, they are forced to reach the inside of the acorn through the basal end, which is more difficult to access. It is likely that these two rodent species, which have been consuming acorns longer, have had the negative experience of the bad taste of the tannins [47] at this apical end of the shell, leading them to adapt to opening the acorns from the basal part, despite the increased difficulty of this action.

The Common vole exhibits a different behaviour, opening acorns at the apical end as verified by acorn remains and recordings [37]. The Common vole has not previously used this type of feed. The lack of previous experience leads it to ignore the bad taste of tannins at this apical end since it does not know how the rest of the shell tastes.

It is likely that the Wood mouse and the Algerian mouse, which have been consuming acorns for a long time, have adapted their behaviour to open the acorns at the basal end, in response to the bad taste of the tannins found in the apical part. However, this behaviour favours the plants since it preserves the embryo, at least during the first attack [24]. Therefore, we believe that the relationship of these two rodent species with oak species, which began as acorn predation by adaptation, has shifted towards the mutualism extreme [17,50,51]. This relationship cannot be truly considered total mutualism, but it approaches it, since the rodents benefit the plant. They engage in this behaviour due to the bad taste of the tannins that the plant has at its apical end [46,47]. However, they favour the oak species through this behaviour [44]. It provides the oaks with a benefit since their acorns can germinate after being partly consumed [45]. The rodents of these two species also benefit, as they did when the relationship was one of mere predation.

The Wood mouse has existed in the Iberian Peninsula for longer than the Algerian mouse [25,26]. It has also probably been engaging in this behaviour for longer, which is why it opens fewer acorns on the ‘wrong’ side, the apical end. The Algerian mouse opens more acorns at the apical end than the Wood mouse, as if it had not yet fully acquired this behaviour. However, the number of acorns that it opens at the basal end is still much greater, even though it has been practising this behaviour for less time.

The behaviour of the three rodent species during the handling, transport and storage of acorns is also different. The Common vole does not transport acorns in this experiment. It consumes most of them on the spot where it finds them. A few are scattered and left on the surface. It also refrains from burying acorns. This behaviour may be explained by the fact that, being a newcomer species, it does not know or has not had previous experience with the new predators that may be lurking in this region [38]. Given this lack of prior experience, it eats the acorns directly where they are found, not worrying about the predators that may attack it. The Wood mice and the Algerian mice, having spent more time in the region, have already had negative experiences with predators, especially birds of prey that stalk them from the sky [38]. For this reason, they seek the protection of bushes, stones or grass to avoid being spotted. The primary dispersal of the acorns of oak species is by barochory, falling by gravity from the tree to be deposited on the ground under the tree. The points where the rodents locate the acorns are under the producing trees, which do not allow the proliferation of many plants due to competition. Therefore, there is little protection under the treetops where the acorns are deposited. If experience leads rodents to search for food under the canopy of these trees, birds of prey also have the experience that the mice that they feed on will be concentrated under the trees [38]. For this reason, since they are unprotected when searching for acorns, the rodents of these two species (Wood mouse and Algerian mouse) must transport the acorns as quickly as possible to protected locations where they can process them. This may be one of the reasons why rodents of these two species initiate the transport of acorns and thus their dissemination. It should be understood that transport is an energy expenditure that would be unnecessary unless there was an important reason for this expenditure.

The differences observed between the Common vole and the other two rodent species in the in situ consumption and transport of acorns may be explained by the adaptations and experiences that have been acquired by the Wood mouse and the Algerian mouse thanks to the longer time that they have spent handling acorns in this region. Some authors have proposed other causes preventing on-site consumption. Any acorn characteristic that lengthens the handling time prevents their consumption in situ, including the presence of tannins [5,52,53]. To reduce the tannin concentration, the acorns must be stored for a long time [47].

Differences are also found between the three rodent species about the behaviour that leads acorns to their final destination [54,55]. Wood mice and Algerian mice consume few acorns in situ due to the aforementioned predation risks. However, they also leave few acorns on the surface. The fundamental difference between the two species is that the Wood mouse puts most of the acorns that it processes in caches [56,57]. This behaviour may be due to the widespread reciprocal theft occurring between conspecifics in these rodent societies [57]. To avoid losing the harvest, each specimen makes shallow accumulations of a small number of acorns in the cache. They are shallow to provide immediate food supplies soon without expending a lot of energy burying and digging up acorns [57]. The caches consist of a small number of acorns because, in the case in which they are stolen, only a small part of the hard-earned acorns will be lost [56]. This type of storage favours the dispersal process because, if the stored acorns germinate, they have little difficulty in emerging [55]. This rodent species participates intensively in the process of acorn dissemination. On the other hand, the Algerian mice accumulate a greater number of acorns in the artificial containers created to mimic their larders [23,43,55]. This behaviour appears to be more typical of rodents that expect to survive the adverse conditions of winter [9]. They accumulate numerous resources so that they do not need to forage during unfavourable winter days [55,58]. This type of storage is deeper than the caches, and if the stored acorns germinate, it is more difficult for them to emerge. The number of acorns stored is also greater than in caches, but the probability of forgetting their location is lower than in the case of caches due to their smaller number and because they are usually resting places, sharing the function of storage and refuge. This species also participates in the acorn dispersal process. The Common vole species does not participate in the dissemination of acorns, firstly because it does not transport acorns and secondly because it behaves as a predator, destroying the acorn embryo. It is a hostile species in terms of the acorn dispersal process. The Common vole has not changed its acorn hoarding behaviour as it does not cache, as we have observed in previous studies.

## 5. Conclusions

The three rodent species having similar characteristics occupy the same territory, but they separate their niches and habitats to avoid the strong competition for resources that would lead to the exclusion of one of them.

The presence of rodent predators in the region is one of the factors determining the choice of habitat for protection.

The presence of these rodent predators may be one of the factors involved in the acorn dispersal process, since it may be one of the causes for the transport of acorns by rodents at the initial stage of the dissemination process. Rodents must seek refuge from predators and therefore transport the acorns to protected places for processing.

The amount of time that a relationship is maintained may lead to adaptations varying the outcome of the relationship. The two species that have been consuming acorns for the longest time have adapted their acorn consumption behaviour by opening them at the basal end, possibly due to prior unpleasant experiences with the taste of the tannins. This behaviour preserves the embryo and serves the interests of the plants by allowing their seeds to germinate. It shifts the relationship from initial predation to mutualism.

The Common vole does not store or transport acorns. It destroys the acorn embryo, behaving as a predator. In this way, it is hostile to the acorn dispersal process. The other two rodent species store acorns. The Wood mouse stores them in caches, favouring germination and the emergence of acorns. The Algerian mouse stores them in larders, possibly hindering the emergence of acorns. These two species collaborate in the acorn dispersal process, exhibiting a relationship that is more similar to mutualism.

The Common vole is not occupying oak forests at the moment, but it may be adapting to the consumption of acorns for their high nutritional value.

## Figures and Tables

**Figure 1 animals-15-01874-f001:**
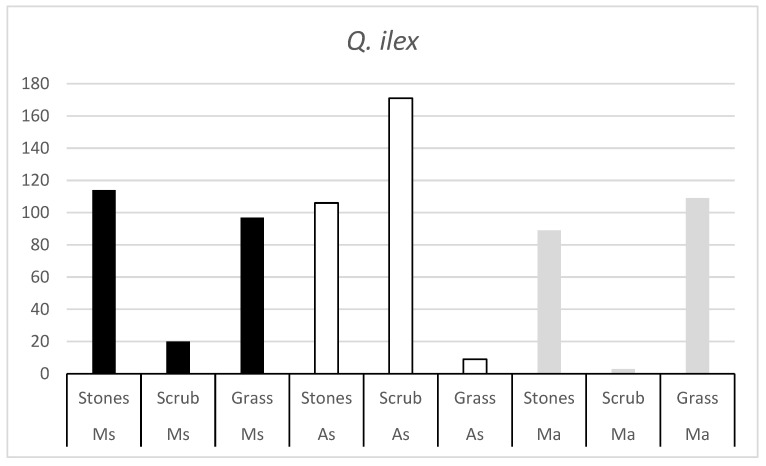
Number of *Q. ilex* acorns consumed by the three rodent species (Ms = Algerian mouse; As = Wood mouse; Ma = Common vole) in the three habitats (stones, scrub and grass).

**Figure 2 animals-15-01874-f002:**
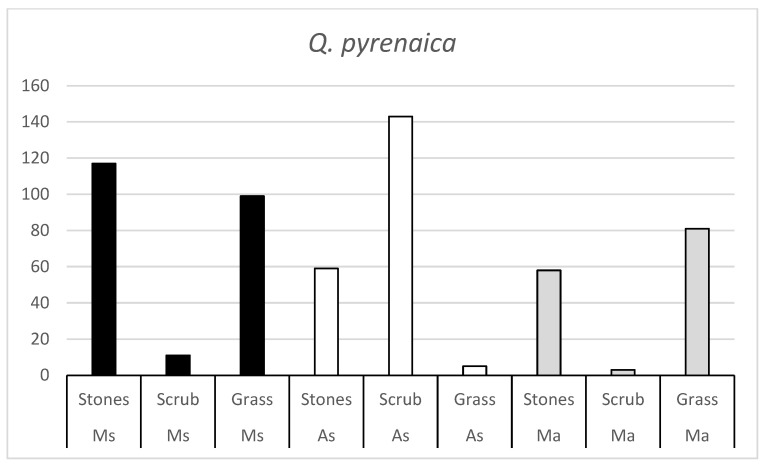
Number of *Q. pyrenaica* acorns consumed by the three rodent species (Ms = Algerian mouse; As = Wood mouse; Ma = Common vole) in the three habitats (stones, scrub and grass).

**Figure 3 animals-15-01874-f003:**
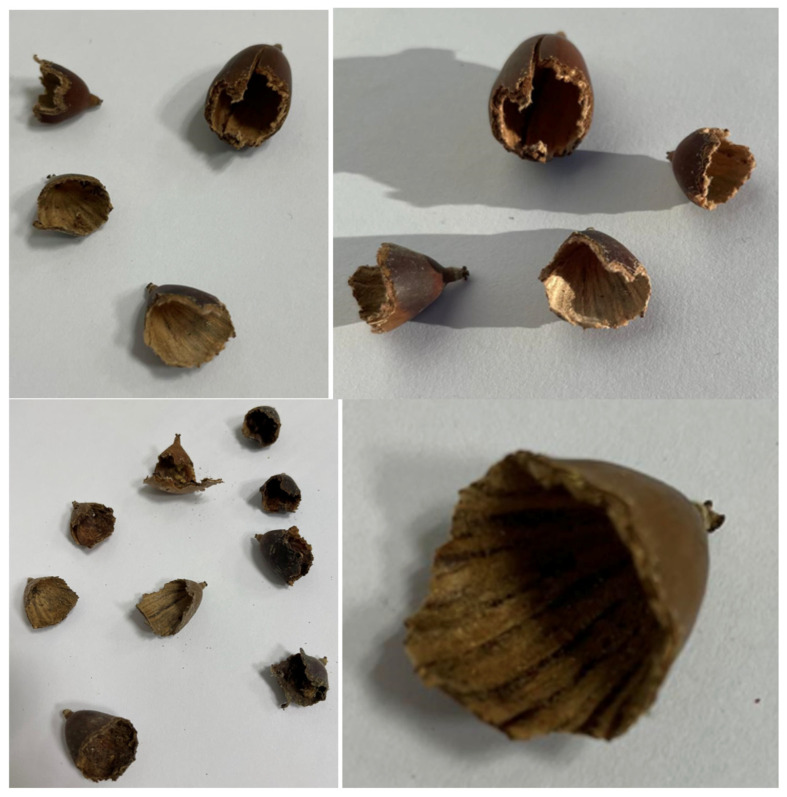
Remains of fully eaten acorn shells showing empty apical dome.

**Table 1 animals-15-01874-t001:** Summary results of linear mixed models testing the effects of rodent and oak species, habitats and time (day) and their interaction on the number of acorns consumed by rodent species. The *F* values of the fixed factors and their significance (*p*) are presented.

	df	*F*	*p*
Number of Acorns	Transported		
Intercept	102	300.2036	<0.0001
*Quercus* species	102	30.2220	<0.0001
Rodent species	102	27.6905	<0.0001
Habitat	102	44.7499	<0.0001
*Quercus* * Rodent	102	6.7821	0.0017
*Quercus* * Habitat	102	2.6364	0.0765
Rodent * Habitat	102	268.8115	<0.0001
*Quercus* * Rodent * Habitat	102	4.0842	0.0041
*Residuals*	*123*		

**Table 2 animals-15-01874-t002:** Summary results of linear mixed models testing the effects of rodent and oak species, habitats and time (day) and their interaction on the mass of acorns consumed by rodent species. The *F* values of the fixed factors and their significance (*p*) are presented.

	df	*F*	*p*
Number of Acorns	Transported		
Intercept	1255	3652.596	<0.0001
*Quercus* species	1255	75.480	<0.0001
Rodent species	1255	687.617	<0.0001
Habitat	1255	0.589	0.5551
*Quercus* * Rodent	1255	12.242	<0.0001
*Quercus* * Habitat	1255	3.994	0.0187
Rodent * Habitat	1255	4.058	0.0028
*Quercus* * Rodent * Habitat	1255	1.686	0.1509
*Residuals*	*1261*		

**Table 3 animals-15-01874-t003:** Mean ± standard error of the mass of acorns consumed per acorn per day by the three rodent species, by the two oak species and in the rodent-oak interaction. Different letters indicate significant differences (*p* < 0.05). 1-way Rodents (ANOVA F2,1273 = 629.67, *p* = 0.00). One-way Oaks (ANOVA F1,1275 = 51.13, *p* = 0.00000). Two-ways Rodents-Oaks (ANOVA F2,1270 = 12.42, *p* = 0.000005).

		n	Weight (g) ± SE
Algerian mouse		454	0.7217 ± 0.0225 a
Wood mouse		492	1.3175 ± 0.0219 b
Common vole		347	1.8726 ± 0.0191 c
*Q. ilex*		729	1.3616 ± 0.0247 d
*Q. pyrenaica*		564	1.1098 ± 0.0243 e
*Algerian mouse*	*Q. ilex*	236	0.8258 ± 0.0351 f
*Algerian mouse*	*Q. pyrenaica*	218	0.6090 ± 0.0255 g
*Wood mouse*	*Q. ilex*	288	1.4559 ± 0.0327 h
*Wood mouse*	*Q. pyrenaica*	204	1.1223 ± 0.0185 i
*Common vole*	*Q. ilex*	191	1.8816 ± 0.0300 j
*Common vole*	*Q. pyrenaica*	142	1.8606 ± 0.0199 j

**Table 4 animals-15-01874-t004:** Acorn removal rate. The values represent the slope of the straight line adjusted to the number of acorns that remain unconsumed each day.

** *Q. ilex* **	**Habitat**	**Slope**
Algerian mouse	Stones	16.821
Algerian mouse	Scrub	2.7857
Algerian mouse	Grass	13.714
Wood mouse	Stones	15.927
Wood mouse	Scrub	25.226
Wood mouse	Grass	1.3452
Common vole	Stones	13.202
Common vole	Scrub	0.4405
Common vole	Grass	16.774
** *Q. pyrenaica* **	**Habitat**	**Slope**
Algerian mouse	Stones	16.964
Algerian mouse	Scrub	1.952
Algerian mouse	Grass	14.31
Wood mouse	Stones	8.8214
Wood mouse	Scrub	21.167
Wood mouse	Grass	0.8333
Common vole	Stones	8.881
Common vole	Scrub	0.4762
Common vole	Grass	11.976

**Table 5 animals-15-01874-t005:** Average weight of acorn mass consumed per acorn per day by the three rodent species in each of the three habitats. Different letters indicate significant differences (*p* < 0.05). One-way Habitat (ANOVA F2,1273 = 2.438, *p* = 0.087745). Two-way Rodent-Habitat (ANOVA F4,1266 = 2.6, *p* = 0.034709).

		n	Weight (g) ± SE
	Stones	541	1.20734 ± 0.0282 a
	Scrub	353	1.2641 ± 0.0276 a
	Grass	385	1.2989 ± 0.0357 a
Algerian mouse	Stones	242	0.7510 ± 0.0338 b
Algerian mouse	Scrub	31	0.5926 ± 0.0801 b
Algerian mouse	Grass	181	0.7045 ± 0.0307 b
Wood mouse	Stones	163	1.3316 ± 0.0401 c
Wood mouse	Scrub	316	1.3229 ± 0.0268 c
Wood mouse	Grass	13	1.0106 ± 0.0692 b
Common vole	Stones	136	1.8701 ± 0.0244 d
Common vole	Scrub	6	1.6377 ± 0.0231 c
Common vole	Grass	191	1.8818 ± 0.0276 d

**Table 6 animals-15-01874-t006:** Number and percentage of *Q. ilex* acorns consumed in different ways and the fate of the embryo: Acorns that retain the embryo (with embryo) and those that do not retain the embryo (without embryo).

				Apical		Apical	
*Q. ilex*		Basal	Intact	with Embryo		Without Embryo	Total
Algerian mouse	Nº	148	240	11	Nº	32	20
	%	33	53	2	%	7	4
		with embryo				Without embryo	
Available	Nº	399			Nº	52	
450	%	89			%	11	
		Attacked	with embryo			Without embryo	Attacked
	Nº	211	159		Nº	52	211
	%		75		%	25	
				Apical		Apical	
*Q. ilex*		Basal	Intact	with Embryo		Without embryo	Total
Wood mouse	Nº	150	236	3	Nº	7	55
	%	33	52	1	%	2	12
		with embryo				Without embryo	
Available	Nº	389			Nº	62	
450	%	86			%	14	
		Attacked	with embryo			Without embryo	Attacked
	Nº	215	153		Nº	62	215
	%		71		%	29	
				Apical		Apical	
*Q. ilex*		Basal	Intact	with Embryo		Without embryo	Total
Common vole	Nº	5	334	0	Nº	50	61
	%	1	74	0	%	11	14
		with embryo				Without embryo	
Available	Nº	339			Nº	111	
450	%	75			%	25	
		Attacked	with embryo			Without embryo	Attacked
	Nº	116	5		Nº	111	116
	%		4		%	96	

**Table 7 animals-15-01874-t007:** Number and percentage of *Q. pyrenaica* acorns consumed in different ways and fate of the embryo: Acorns that retain the embryo (with embryo) and those that do not retain the embryo (without embryo).

				Apical		Apical	
*Q. pyrenaica*		Basal	Intact	with Embryo		Without Embryo	Total
Algerian mouse	N°	142	262	5	N°	22	19
	%	32	58	1	%	5	4
		with embryo				Without embryo	
Available	N°	409			N°	41	
450	%	91			%	9	
		Attacked	with embryo			Without embryo	Attacked
	N°	188	147		N°	41	188
	%		78		%	22	
				Apical		Apical	
*Q. pyrenaica*		Basal	Intact	with Embryo		Without Embryo	Total
Wood mouse	N°	107	301	1	N°	5	36
	%	24	67	0	%	1	8
		with embryo				Without embryo	
Available	N°	409			N°	41	
450	%	91			%	9	
		Attacked	with embryo			Without embryo	Attacked
	N°	149	108		N°	41	149
	%		72		%	28	
				Apical		Apical	
*Q. pyrenaica*		Basal	Intact	with Embryo		Without Embryo	Total
Common vole	N°	6	348	0	N°	56	40
	%	1	77	0	%	12	9
		with embryo				Without embryo	
Available	N°	354			N°	96	
450	%	79			%	21	
		Attacked	with embryo			Without embryo	Attacked
	N°	102	6		N°	96	102
	%		6		%	94	

**Table 8 animals-15-01874-t008:** Fate of *Q. ilex* acorns after being transported or *used* by rodents of each species (Algerian mouse, Wood mouse, Common vole).

Acorn Fate					
*Q. ilex*	Algerian Mouse				
HABITAT	In Situ	Surface	Cache	Larder	Habitat fate
STONES	3	14	24	61	Stones
SCRUB	1	6	1	2	Scrub
			3	3	Stones
			1	2	Grass
GRASS	3	13	23	41	Grass
			2	7	Stones
					Scrub
*Q. ilex*	Wood mouse				
HABITAT	In Situ	Surface	Cache	Larder	Habitat fate
STONES			21	16	Stones
		9	19	14	Scrub
SCRUB	4	10	55	36	Scrub
			12	9	Stones
GRASS	1	3			Grass
				1	Stones
			3	1	Scrub
*Q. ilex*	Common vole				
HABITAT	In Situ	Surface	Cache	Larder	Habitat fate
STONES	34	7		5	Stones
SCRUB	3				Scrub
GRASS	46	9		9	Grass

**Table 9 animals-15-01874-t009:** Fate of *Q. pyrenaica* acorns after being transported or *used* by rodents of each species (Algerian mouse, Wood mouse, Common vole).

Fate of Acorns					
*Q. pyrenaica*	Algerian Mouse				
HABITAT	In Situ	Surface	Cache	Larder	Habitat fate
STONES	2	9	23	58	Stones
SCRUB	3		1	2	Scrub
			1	4	Stones
					Grass
GRASS	2	5	20	41	Grass
		4	2	7	Stones
		1	1	1	Scrub
*Q. pyrenaica*	Wood mouse				
HABITAT	In Situ	Surface	Cache	Larder	Habitat fate
STONES	2	1	9	3	Stones
		3	16	5	Scrub
SCRUB	3	6	53	24	Scrub
		3	11	5	Stones
GRASS	2				Grass
					Stones
			2	1	Scrub
*Q. pyrenaica*	Common vole				
HABITAT	In Situ	Surface	Cache	Larder	Habitat fate
STONES	31	9		3	Stones
SCRUB	2	1			Scrub
GRASS	43	8		5	Grass

## Data Availability

The datasets generated during the current study are available in the [https://uvadoc.uva.es/handle/10324/75626] repository accessed on 1 January 2020.

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
