# Peer review of "How Rodent Species Adapt to the Food Resources of Their Habitat"

_animals, 2025, doi:10.3390/ani15131874_

Round 1
Reviewer 1 Report
Comments and Suggestions for Authors
The aims of this research are clear but the authors are based on an erroneous statement: The Line 41 is not true, due to the common vole has been found in the central Iberian Peninsula for over 400 ka, where it has remained until the present day. Specifically, in the northern Iberian sub-plateau, there are numerous records of this vole at sites in the Atapuerca mountain range (Burgos) in chronologies spanning from 400 ka to the Neolithic and Bronze Ages, and also in the province of Segovia at sites from the Late Pleistocene (Alfaro-Ibáñez and Cuenca-Bescós, 2023; Bañuls and Bisbal-Chinesta, 2022; Cuenca-Bescós et al., 2001, 2016; Sala et al., 2020 among others)
Alfaro-Ibáñez, M. P., Cuenca-Bescós,G. (2023). New small mammal species identified in the historical site of the Valdegova cave (Burgos, Spain). 4th Virtual Palaeontological Congress, Book of Abstracts, 136.
Bañuls-Cardona, S., Bisbal-Chinesta, J. F. (2022). Small Vertebrate Accumulations from El Mirador Cave: A Climate and Ecological Analysis. Allué, E., Martín, P., Vergès, J.M. (eds) Prehistoric Herders and Farmers. Interdisciplinary Contributions to Archaeology. Springer, Cham.
Cuenca-Bescós, G. and Laplana, C. (2001). La séquence des rongeurs (Mammalia) des sites du Pléistocène inférieur et moyen d'Atapuerca (Burgos, Espagne). L'Anthropologie, 105:115-130.
Cuenca-Bescós, G., Blain, H.-A., Rofes, J., López-García, J. M., Lozano-Fernández, I., Galán, J., Núñez-Lahuerta, C. (2016). Updated Atapuerca biostratigraphy: Small-mammal distribution and its implications for the biochronology of the Quaternary in Spain. Comptes Rendus Palevol, 15 (6):621-634.
Sala, N., Pablos, A., Gómez-Olivencia, A., Sanz, A., Villalba, M., Pantoja-Pérez, A., Laplana, C., Arsuaga, J. L., Algaba, M. (2020). Central Iberia in the middle MIS 3. Paleoecological inferences during the period 34-40 cal kyr BP. Quaternary Science Reviews, 228:106027.
The aim of this research is the understanding how three rodents (Apodemus sylvaticus, Mus spretus, Microtus arvalis) have adapted to the conditions of the occupied environment, comparing their behavior and the use of resources and in the occupation of habitats.
The authors want to infer the relationship between rodents and oaks in relation with the time that they have lived in an area.
Two of the three species (which lived in the Iberian Peninsula during more time) that have consumed oaks during more time may have acquired adaptations that the other rodent hasn’t it. The research carried out includes an experiment with the three rodents living in the Iberian Peninsula in different habitats using acorns (from Q. ilex and Q. faginea) as food resource.
Nevertheless, as one of the authors of this manuscript acknowledge in previous work, "the diet of this species" - the common vole - "includes fresh green plants, but not acorns" (Del Arco et al., 2018, p. 2). Therefore, it is meaningless to compare this species' acorn-processing behavior with that of other species if the common vole does not consume them.
Line 96: If this is the objective of this study, it would be necessary to consider the time from which each species selected in this study has lived in environments with acorn availability, regardless of their early or late arrival in the geographic region considered in this study. For example, suppose a species not present in the study area originates from other areas where Quercus (and acorns) are present. In that case, it should be considered that this species has a long temporal relationship with this food resource.
The common vole has been present in the region considered (the northern sub-plateau) from 400 ka until the end of the Holocene. During this time, different Quercus species have been recorded in the form of pollen grains in the sites where this rodent species is found (Expósito et al., 2017; López-García et al., 2010; Rodríguez et al., 2011, suppl. data 2; among others). Therefore, the common vole has a long history in common with the acorn.
Expósito, I., Burjachs, F., Vergès, J. M. 2017. Human trace on the landscape during the Holocene at El Mirador Cave (Sierra de Atapuerca, Spain): The palynological evidence. The Holocene 27(8): 1201-1213.
López-García, J. M., Blain, H.-A., Cuenca-Bescós, G., Ruiz.Zapata, M. B., Dorado-Valiño, M., Gil-García, M. J., Valdeolmillos, A., Ortega, A. I., Carretero, J. M., Arsuaga, J. L., Bermúdez de Castro, J. M., Carbonell, E. (2010). Palaeoenvironmental and paleoclimatic reconstruction of the Latest Pleistocene of El Portalón Site, Sierra de Atapuerca, northwestern Spain. Palaeogeography, Palaeoclimatology, Palaeoecology, 292:453-464.
Rodríguez, J., Burjachs, F., Cuenca-Bescós, G., García, N., van der Made, J., Pérez González, A., Blain, H.-A., Expósito, I., López-García, J. M., García Antón, M., Allué, E., Cáceres, I., Huguet, R., Mosquera, M., Ollé, A., Rosell, J., Parés, J. M., Rodríguez, X. P., Díez, J. C., Rofes, J., Sala, R., Saldié, P., Vallverdú, J., Bennasar, M. L., Blasco, R., Bermúdez de Castro, J. M., Carbonell, E. (2011). One million years of cultural evolution in a stable environment at Atapuerca (Burgos, Spain). Quaternary Science Reviews, 30:1396-1412.
In the introduction I can suggest some comments:
In the methodology the authors indicate two types of Quercus (line 125): Q. ilex and Q, faginea, that use for the experiment. Nevertheless, in the line 138 the authors indicate Q. ilex and Q. pyrenaica. Please, let me know which species are the correct. Thank you.
2.2. Characteristics of species
Line 160: The following reference is also relevant and clarifies the chronology of the entry of this species into the Iberian Peninsula; please add it:
Domínguez García, A. C., Laplana, C., Sevilla, P., Blain, H.-A., Palomares Zumajo, N., Benítez de Lugo Enrich, L. (2019). New data on the introduction and dispersal process of small mammals in southwestern Europe during the Holocene: Castillejo del Bonete site (southeastern Spain). Quaternary Science Reviews, 225:1.
Line 181: There is no evidence of acorn consumption by the common vole except in captivity. In previous work, the authors of this manuscript indicate that the common vole does not consume acorns. Why consider this species in this hypothetical situation?
I believe the authors' research is founded on two incorrect assumptions.
- The common vole arrived on the central plateau of the Iberian Peninsula in the 80s of the last centuries (Line 169, 170). Nevertheless, it has been found in the central Iberian Peninsula for over 400 ka and the authors don’t consider it in this study.
- The common vole diet based on acorn, that it is not true.
So, please, I think the research's aims must be changed and reconsidered to indicate why the authors conducted this hypothetical experiment when this real situation is almost impossible.

Author Response
Following your instructions, we are sending a new document which has undergone an exhaustive linguistic revision.

Reviewer 2 Report
Comments and Suggestions for Authors
The main problem of the manuscript is the following:
The main hypothesis authors tested, is that the depth of co-adaptation of rodents and their food (acorns of the two oak species) depend on the duration of the rodent-oak relationship. The results of experiments presented are considered by authors as supporting their hypothesis. However, it seems correct for only two of three rodent species (Apodemus sylvaticus and Mus spretus). Concerning the third species (Microtus arvalis), it is obviously not so. Microtus arvalis is a folivorous species with deep morphological and physiological adaptation to feed on green leaves of grasses. The main direction of evolution of digestive system in rodents is from omnivorous and/or granivorous type of diet to folivorous diet, and reverse evolution seems impossible due to morphological and physiological constraints. Thus, low efficiency of foraging Microtus arvalis on oak acorns seems to be the result of its morpho-physiological adaptation to folivorous diet, but of its recent arrival in the study area.
The English of the manuscript should be significantly improved.
Some minor notes:
In the Abstract, please add the names of studied species.
In the Results, please, explain better what are the differences between analyses presented in tables 1 and 2.
Comments on the Quality of English LanguageThe English of the manuscript should be significantly improved.
Author Response
Following your instructions, we are sending a document that has undergone an exhaustive linguistic revision.

Reviewer 3 Report
Comments and Suggestions for Authors
The manuscript titled “How rodent species adapt to the food resources of their habitat” presents an ecologically relevant study examining the interactions between three rodent species and two oak species, with a focus on acorn consumption, handling behavior, and habitat segregation. The central hypothesis that prolonged co-existence drives rodent-oak interactions from antagonism toward mutualism is compelling and well-motivated. However, the manuscript needs following corrections
Major Comments:
The phrase “several specimens” should be quantified. Specify the number of individuals captured per species, as the study likely involved a fixed number. Additionally, clarify the capture method.
The manuscript should include a detailed description of the experimental design and clearly state the assumptions tested (e.g., normality, homoscedasticity, independence).
Report the total sample size. Tables 1 and 2 lack key model outputs such as effect sizes and confidence intervals.
The use of lab-reared juvenile rodents in artificial enclosures with manipulated microhabitats raises concerns about ecological validity. The assumption that naive rodents accurately reflect natural behavior is questionable. Clarify whether environmental variability (e.g., presence of predators, seasonal effects) was adequately represented.
The conclusion that habitat segregation prevents competition lacks strong data. The observed partitioning might be coincidental or context-dependent. Also, the manuscript does not address interspecific interactions (e.g., aggression, overlap), which are essential to support this claim.
Minor Comments:
Figures 1 and 2: These are not labeled or described in the text.
Line 12: “began as predation” → consider rephrasing as “initially characterized by predation.”
Line 18: “due to brutal competition” → replace “brutal” with a more neutral term such as “intense” or “strong.”
Line 24: Awkward phrasing: “We want to know if time is the origin of adaptations…” revise it
Line 31: Typo: “destroy them during consumption” → should be “destroys them during consumption.”
Lines 46–47: Citation format error remove the extra closing parenthesis in “[4]).”
Lines 96–98: Rephrase: “The objective we want to achieve is to test…” → “The objective is to determine…”
Line 122: “Avoid aggressive confrontations” → consider adding a citation or explanation regarding aggression rates among rodents under similar conditions.
Line 137: Clarify the rationale for using four cage openings (e.g., to simulate natural foraging conditions?).
Line 234 (Fig. 1): Consider adding error bars or statistical annotations to indicate significance.
Line 260: Typo: “table 3” → capitalize to “Table 3.”
Author Response

(The authors gave the same response as above.)

Reviewer 4 Report
Comments and Suggestions for Authors
Thank you for the opportunity to review this manuscript. The authors evaluate relationships of 3 rodents to acorn consumption patterns and acorn dissemination in a captive setting. In general, the manuscript could be tightened up considerably to aid in the reader’s understanding. Some specific recommendations to achieve this are found below. Additionally, while the manuscript focuses on animal adaptations it is not clear whether the experiment was conducted over a large enough period time to clearly demonstrate animal adaptation. Consider refocusing the manuscript from species adaptations to differing foraging strategies of the species – this too will aid in streamlining the presentation and assist with the reader’s understanding.
Specific:
Line 1 – Was adaptation demonstrated in this manuscript?
Line 11 – Simple summary mentions time, which is paramount to animal adaptation. When were the experiments conducted? For how long?
Line 23 – “just arrived in the area” In the experiments described, the 3 species all just arrived in the experimental area. It is not clear to the reader when experiments were conducted or for how long they occurred. Recommend additional information on the temporal components of experiments.
Line 39 – Recommend providing scientific names here.
Lines 48-51 – This sentence reads like Methods, not introduction material. Recommend rewording.
Lines 93-95 – Again, these sentences are more Methods than Introduction. Recommend rewording.
Line 96-108 – Recommend a more straightforward and concise presentation of objectives. Simply state, “Our objectives were to 1)…2)…and, 3). This will provide clarity for the reader.
Lines 112-113 – Recommend removing scientific names here.
Line 144 – “Every day” Strongly recommend including more details on temporal aspects of experiments. As written, the experiments are not repeatable. For how long? Was this a long enough duration to capture adaptation?
Line 144 – How were mobilized acorns tracked? PIT tags, color-coded, etc. More details are needed on how acorn movement was tracked.
Line 155 – This entire subheading seems out of place, as rodent characteristics are not Methods. For clarity, recommend restructuring manuscript.
Line 202 – What aspects of time were used in analyses? More details are needed for reader’s understanding.
Line 218-221 – This sentence is redundant with Table caption. Recommend deleting this from text and allowing the Table to speak for itself.
Line 240-241 – This sentence is redundant with Figure caption. Recommend deleting this from text and allowing the Figure to speak for itself.
Line 247-249 – This sentence is redundant with Table caption. Recommend deleting this from text and allowing the Table to speak for itself.
Line 310 – This sentence is redundant with Table caption. Recommend deleting this from text and allowing the Table to speak for itself.
Line 345-355 – This sentence is redundant with Table caption. Recommend deleting this from text and allowing the Table to speak for itself.
Line 479 – “In a conscious way? We think not”. Recommend rewording these sentences as they seem out of place for a manuscript reporting science. Throughout there are instances where presentation is far too informal for scientific presentation; this results in unnecessary words and writing that is not clear or concise.
Line 498-500 – Were any of the experimental animals consumed by raptors during experiments?
Line 564 – This is my main criticism of the manuscript related to adaptation. Common voles do not store acorns and do not in your experiments. Did you expect them suddenly start caching? Adaptation could be demonstrated if over time they started caching acorns, but it is not clear if the experiments were conducted over a large enough time to capture this. Again, recommend refocusing the manuscript away from adaption and toward foraging strategies/behaviors.
Thanks again for the opportunity.
Author Response

(The authors gave the same response as above.)

Round 2
Reviewer 1 Report
Comments and Suggestions for Authors
We appreciate the authors' efforts in responding to our comments and making changes to the manuscript, which has improved compared to the initial version. However, the authors highlight one crucial issue in their responses to our comments, which must be reflected in the manuscript.
In their response to our comments on the manuscript, the authors indicate that: “Currently in the study area it (Microtus arvalis) does not consume them (acorns) because it has not come to occupy the oak forests that produce this seed. This is the risk we see to the presence of this species moving from crops and meadows to these forests…” (last paragraph of page 2 of the authors' responses to reviewer 1's comments).
This last statement (in bold) is not included in the manuscript. It serves to justify the analysis of this rodent's consumption of a resource, acorns, for which there was no evidence until now. The authors should incorporate this statement and supporting bibliographic citations in section 2.2 (Species characteristics), documenting that Microtus arvalis is shifting its habitat from crops and grasslands to oak forests in the studied area. Consequently, this rodent is beginning to utilise a resource that was previously unavailable to it. This point is critical, as it provides essential context for the rest of the manuscript, which would otherwise seem meaningless.
If Microtus arvalis is not currently transitioning from grasslands and crops to forested areas in the study region, then its consumption of acorns would be purely hypothetical. While this behaviour may have been observed in captivity, it has not been documented in the wild. This distinction is significant because, if this is the case, the species cannot be classified as an acorn predator, as it does not consume acorns in its natural habitat.
Author Response
Reviewer 1
We are grateful for this reviewer's indication. He is right that we had not expressed ourselves clearly with this crucial aspect for the outcome of the experiment. This has led us to introduce a new approach that was already among the results of the experiment, but which we had not been able to interpret clearly. The grounds on which we base our assertion that this species may pose a risk to acorn dispersal processes are personal communications from colleagues working with this species in the study area (Luque-Larena Group). In their experiments they have observed the presence of acorns in the burrows and tunnels that this species makes, but as this is not part of their research objectives they have not published anything about it, so we have no documents to support these assessments. However, with the results of our experiment we can answer these questions. In the experiment in which we placed the rodent species to choose 3 types of habitats, Common vole could choose the scrub habitat. This habitat is constructed within an oak forest in which we have not altered any of its components: adult trees of 3 and 4 metres in height together with younger specimens that form the undergrowth and lesser shrubs that accompany the oak species in their ecosystems. If Common vole chooses this ecosystem, it would indicate that it is beginning to occupy this type of forest. It is then likely that it is starting to use acorns as a food source. In the experiment it has its usual food on the plot, fresh grass, but also acorns. Your choice indicates which food you prefer. Comparing the number and mass of acorns consumed with those used by the other two species, we can see that they are similar in quantity. This indicates that it appreciates them and even prefers them to its own food. Therefore, it is not strange that they are beginning to appear acorns in their burrows because they use them as food, probably because their usual food is scarce in specific periods, during plagues or drought, but we cannot deduce this from our results. On the other hand, we have found that the oak ecosystem is not active, which indicates that it does not occupy them at the moment. There is no record of it entering this type of habitat.
We hope that the modifications introduced will clarify the contributions of this work to the knowledge of these relationships between rodents and oak trees. The modifications made are as follows: in Introduction we have highlighted the importance of this species; we have modified the objectives to include the aspects that refer to the presence of this species; in Material and Methods, in the section on species characteristics we have included the reference to the risk posed by this species; in Discussion we have commented on the results we already had, but focused on the contributions of the possible presence of this species in forests and the use of acorns; and finally we have added a new conclusion to show that this species does not occupy oak forests for the moment, but it is beginning to use acorns as a source of resources.
Reviewer 3 Report
Comments and Suggestions for Authors
The manuscript has been adequately revised, but some corrections are still needed in typography and language.
Author Response
We have carried out a linguistic revision of the document. Please find attached the invoice for this revision

Reviewer 4 Report
Comments and Suggestions for Authors
Manuscript summarized:
In previous work you determined that wood mouse and Algerian mouse have adapted to consume acorns (Del arco et al. 2028). Here you conduct an experiment including these 2 species, plus common vole, which does not consume acorns and has never been presented with them. The 2 former species foraged on acorns, including caching them, as previously noted. The same occurred for common vole - they opened acorns wrong (in inefficiently) and did not cache them. This was expected.
Again, how was adaptation demonstrated in this manuscript?
It was already known that wood mouse and Algerian mouse were adapted to consuming acorns and common vole displayed no signs of adaptation.
There is no new information presented in this manuscript relative to adaptation. Therefore, I (again) recommend refocusing presentation.
With thanks.
Author Response
Reviewer 4
You are right, but the contributions made by other reviewers have allowed us to see that there is more information in our results from which we had not been able to draw the appropriate conclusions. In this sense, we have refocused on objectives of the work.
Common vole arrived from the mountains surrounding the meseta to the northern plateau of the Iberian Peninsula about 45 years ago. It has been done following the expansion of irrigated crops towards the centre of the plateau. Its initial habitat was crops and meadows. But in these 45 years, this species has suffered periods of high density until it has become a plague that devastates crops and meadows until it has exhausted its usual food. This food has also been depleted during prolonged periods of drought in the area. During these periods without food, it is likely that some specimens, given the lack of resources, may have looked for new sources of food. They may have found acorns. In previous studies we have forced this species to consume acorns in the absence of other types of food. The species did not suffer any detriment to its health because acorns are very nutritious. Through personal communications from colleagues working with this species in the same study area (Luque-Larena group) we have been informed that they have begun to detect the presence of acorns in the burrows and tunnels that this species makes. The results are not published because this group does not have among its objectives feeding with acorns. For this reason, we are unable to provide documentation to support this fact. This has led us to think that this species is possibly adapting to new circumstances (lack of food due to pests and drought) in this new occupied area, the central plateau. Through this adaptation, it is possible that it is beginning to occupy the oak forests in the area and is starting to use acorns as a new source of resources. If this is true, the presence of this species in the area could pose a risk to the acorn dispersal process, as it destroys the embryo when it eats them, as we have seen in previous studies. This would mean a drastic reduction in the process of acorn dissemination, which would reduce the possibilities of expansion of these oak forests.
Among the results we have obtained in this work we can check if this species is beginning to occupy oak forests. In the experimental plot we have constructed 3 types of habitats: Stones, Grass (its natural habitat) and Scrub. The scrub habitat is built in an oak forest as indicated in the material and methods section, in which we did not make any modifications. It consists of adult trees of two oak species, juvenile specimens of these species that form the undergrowth and shrubs that accompany these oak species in their habitats. If, once the experiment has been carried out, we find that this species occupies this habitat, it means that it is beginning to occupy a new habitat, the oak forests, in a new adaptation. We have seen that this is not happening, so we conclude that for the moment it is not occupying them.
Another question we can test in this experiment with the results obtained is whether this species is appreciating a new type of food, acorns. In the experiment we have provided acorns, but it also has its usual food, fresh grass, distributed throughout the three habitats offered. As a result, we obtained that Common vole consumes acorns in the same quantity and mass as the other two rodent species, which have been consuming them in their diet for much longer than Common vole. This indicates that it prefers even acorns to its usual food because they are probably more nutritious. Therefore, it is likely, as colleagues have observed, that acorns and their remains are already appearing in their burrows and tunnels because they are beginning to use them as a source of resources in a new adaptation to the newly occupied habitat, the central plateau of the Iberian Peninsula.
We hope that the modifications introduced will clarify the contributions of this work to the knowledge of these relationships between rodents and oak trees. The modifications made are as follows: in Introduction we have highlighted the importance of this species; we have modified the objectives to include the aspects that refer to the presence of this species; in Material and Methods, in the section on species characteristics we have included the reference to the risk posed by this species; in Discussion we have commented on the results we already had, but focused on the contributions of the possible presence of this species in forests and the use of acorns; and finally we have added a new conclusion to show that this species does not occupy oak forests for the moment, but it is beginning to use acorns as a source of resources.